# Arsenic: A Perspective on Its Effect on Pioglitazone Bioavailability

**DOI:** 10.3390/ijerph20031901

**Published:** 2023-01-20

**Authors:** María Cruz del Rocío Terrones-Gurrola, Patricia Ponce-Peña, José Manuel Salas-Pacheco, Abelardo Camacho-Luis, Amaury de Jesús Pozos-Guillén, Guillermo Nieto-Delgado, Olga Dania López-Guzmán, Angel Antonio Vértiz-Hernández

**Affiliations:** 1Coordinación Académica Región Altiplano, Universidad Autónoma de San Luis Potosí, Matehuala 78700, Mexico; 2Facultad de Ciencias Químicas, Universidad Juárez del Estado de Durango, Durango 34120, Mexico; 3Instituto de Investigaciones Científicas, Universidad Juárez del Estado de Durang, Durango 34000, Mexico; 4Centro de Investigación en Alimentos y Nutrición, Facultad de Medicina y Nutrición, Universidad Juárez del Estado de Durang, Durango 34000, Mexico; 5Facultad de Estomatología, Universidad Autónoma de San Luis Potosí, San Luis Potosí 78290, Mexico; 6Departamento de Físico-Matemáticas, Universidad Autónoma de San Luis Potosí, San Luis Potosí 78290, Mexico

**Keywords:** pioglitazone, CYP 450, arsenic, iron, metabolism

## Abstract

Arsenic (As) is a common contaminant in drinking water in northeastern Mexico, which reduces the expression of cytochrome P450 (CYP 450). This enzyme group metabolizes numerous drugs, such as oral antidiabetic drugs such as pioglitazone (61% CYP 3A4, 49% CYP 2C8). When CYP 450’s function is inadequate, it has decreased therapeutic activity in type 2 diabetes mellitus (T2DM). This study aimed to establish the effect of As on pioglitazone metabolism in patients with T2DM. Methodology: Urine, water, and plasma samples from a healthy population (*n* = 11) and a population with T2DM (*n* = 20) were obtained. Samples were analyzed by fluorescence spectroscopy/hydride generation (As) and HPLC (pioglitazone). Additionally, CYP 3A4 and CYP 2C8 were studied by density functional theory (DFT). Results: The healthy and T2DM groups were exposed via drinking water to >0.010 ppm, Ka values with a factor of 4.7 higher, Cl 1.42 lower, and ABCt 1.26 times higher concerning the healthy group. In silico analysis (DFT) of CYP 3A4 and CYP 2C8 isoforms showed the substitution of the iron atom by As in the active sites of the enzymes. Conclusions: The results indicate that the substitution of Fe for As modifies the enzymatic function of CYP 3A4 and CYP 2C8 isoforms, altering the metabolic process of CYP 2D6 and CYP 3A4 in patients with T2DM. Consequently, the variation in metabolism alters the bioavailability of pioglitazone and the expected final effect.

## 1. Introduction

Arsenic (As), in its natural form, is found in the Earth’s crust and is distributed through water in the environment [1]. Its inorganic compounds contaminate water and foodstuffs such as rice, cereals, and livestock [2,3]. It can be found as a soluble species in two oxidation states: As^3^+ (arsenite) and As^5^+ (arsenate), and less frequently as As^0^ (elemental As) and As^3^- (arsine) [4]. Human exposure can be dermal, respiratory, or oral via water [5,6]. After ingestion, As^5^+ is reduced to As^3^+, followed by methylation and reduction, forming methylated and dimethylated As compounds of both oxidation states [7,8,9].

As^3^+ and As^5^+ compounds are mobile in the environment, with As^3^+ being more labile and toxic to most life forms [10]. Chronic exposure to these compounds via drinking water is a health problem affecting >130 million people daily in approximately 35 countries [11,12]. In regions where groundwater is the source of drinking water, exposure to inorganic As has been associated with cardiovascular disease, liver and gallbladder cancer [13,14], neurological disorders, peripheral neuropathies, muscle dysfunction [15], eye disease, skin lesions, and type 2 diabetes mellitus (T2DM) [16,17]. In the Americas, the numbers of people living with T2DM have tripled since 1980, reaching 62 million, estimated to reach 109 million in 2040 [18]. This metabolic disorder is characterized by hyperglycemia, insulin resistance in peripheral tissues, and altered pancreatic beta (β-pancreatic) cell secretory capacity [19]. Glucose levels determine the glycosylated hemoglobin (Hb) percentage in the blood (HbAc1α). The level for a diagnosis of T2DM is 6.5% or higher; for prediabetes, between 5.7% and 6.4%, and <5.7% indicates no diabetes [20]. HbAc1α results from non-enzymatic glycosylation of Hb, reflecting blood glucose levels 3–4 months before testing [13]. One estimate cites 346 million people have diabetes, 90% of whom have T2DM. Likewise, in residents of countries such as Bangladesh, Taiwan, and Mexico, chronically exposed to high concentrations of inorganic As in drinking water (>100 µg/L), an increase in the incidence and prevalence of T2DM has been observed, despite the treatments applied [21].

For this condition, pharmacological therapies include a low-carbohydrate diet, lifestyle changes, and drug treatment based on insulin and oral hypoglycemic agents. Of the latter, pioglitazone from the thiazolidinedione group is the most widely used, which binds to PPARs and favors gene transcription whose products affect carbohydrate and lipid metabolism. In addition, it tends to control glycemia due to its insulin-sensitizing effect, mediated by its action on PPAR-γ 1 and 2 [22].

On the other hand, liver isoenzymes CYP 3A4 and CYP 2C8 from cytochrome P450 (CYP 450) are involved in 52% of drug metabolism, such as of pioglitazone [23]. CYP 450 constitutes the principal family of lipophilic drugs and xenobiotic catalytic enzymes [24,25,26]. There are eight families and forty-four subfamilies, where the control is multifactorial. It is due to additional polymorphisms in regulatory transgenes and non-genetic host factors (sex, age, disease, hormonal influence, diurnal influences, and other factors). Isoenzymes are relevant in pharmacogenetics and environmental toxicology because they can evaluate drug interactions, drugs with environmental xenobiotics, and other endogenous metabolism relations [27,28].

CYP 450 enzymes are hemoproteins with 420 to 503 amino acids. They have a thiol group (-SH), an iron atom (Fe++; heme group), and C-terminal and N-terminal [28]. CYP 3A4 and CYP 2C8 enzymes are most involved in drug metabolism (52% of metabolized drugs) [23]. However, contaminants, such as those at concentrations of 2.5–5 µM, favor a decrease in CYP 3A4 expression [29]. At 40 ppb in drinking water in mice, the expression of CYP 1A1/2, CYP 2B, CYP 3A4, and CYP 3A23 is negatively regulated [30]. Therefore, CYP 3A4 and CYP 2C8 metabolize pioglitazone, but under As’s presence, they may exhibit variations in the metabolization proportion and changes in its bioavailability. For this reason, this study aimed to determine the effect of As on pioglitazone metabolism, its relation with the patient’s glycemic control under treatment, and its effectiveness in treating patients with T2DM.

## 2. Materials and Methods

### 2.1. Materials and Reagents

Tedia’s ethyl acetate (HPLC grade), JT Baker’s methanol (HPLC grade), JT Baker’s acetonitrile (HPLC grade), CTR acetic acid (reagent grade), JT Baker’s nitric acid (analytical grade), JT Baker’s hydrochloric acid (analytical grade), JT Baker’s sodium hydroxide (analytical grade), JT Baker’s potassium iodide (analytical grade), ALDRICH’S sodium borohydride (analytical grade), SIGMA’s ascorbic acid (analytical grade), triple distilled water, pioglitazone hydrochloride, Zactos 15 milligrams (Eli Lilly), pioglitazone hydrochloride, Zactos 30 milligrams (Lot No. A621450A., manufactured by Eli Lilly, bottle with seven tablets, Registration No. 061M2000SSA IV), reference standard 1643e, National Institute of Standards and Technology (NIST; water standard 60.45 ± 0.72 µgAs/L), reference standard ClinChek Control Urine, Control Lyophilized for Trace Elements Level (IRIS; urine standard 32–51 µgAs/L) were used.

### 2.2. Study Population

Inhabitants of Peñón Blanco, Durango, Mexico participated in the study, divided into clinically healthy volunteers (healthy group; *n* = 11) and volunteers with T2DM (T2DM group; *n* = 20).

Inclusion criteria: (1) people drinking or cooking with municipal drinking water, (2) both sexes, (3) ages between 35 and 65 years, (4) residents of Peñón Blanco, at least 20 years before the study, (5) body mass index (BMI) < 30 Kg/m^2^; for both study groups.

Exclusion criteria: (1) have presented, have suspicion of, or adverse reaction to thiazolidinediones, cardiac complications, degenerative joint disease, renal or hepatic complications, neoplasms, consumption treatment or food that inhibits CIT-P450, pregnancy, tobacco or alcohol consumption; (2) presenting reaction or discomfort during the study, voluntary withdrawal, and deviation from the inclusion criteria.

The study was conducted under the approval of the Ethics Committee of the Faculty of Medicine of the Universidad Juárez del Estado de Durango. All participants provided signed informed consent.

### 2.3. Clinical and Anthropometric Analyses

Plasma glucose, HbAc1α, and insulin in peripheral blood were measured based on the analysis protocols of the Clinical Laboratory of the Scientific Research Institute of the Universidad Juárez del Estado de Durango. 

The anthropometric data recorded corresponded to sex, age, weight, height, systolic blood pressure (SBP), diastolic blood pressure (DBP), and heart rate, all obtained at the time of sampling. The analysis of the anthropometric parameters shows the mean ± standard error. 

### 2.4. Methodology

#### 2.4.1. Arsenic Determination

Urine: The samples were taken from the morning’s first urine in duplicate; with 15 days between each collection, after a 24 h fast of white and red protein, it was suggested that at the beginning of urination, three seconds should elapse before the urine is collected. The urine was delivered in the first hours after collection and subjected to low temperatures (4 °C and then −20 °C) until analysis and the urine density (approximately 1) adjusted.

Each sample was centrifuged in polypropylene tubes for 10 min at 3000 rpm and the sediment was discarded. Then, 1.5 mL of the supernatant was placed in a dry bath (80 °C), 1.5 mL of HNO_3_ was added to each tube, and then the tube was closed. Sixty minutes were timed from the addition of HNO3 to the first tube. Once the time had elapsed, the tubes were removed from the dry bath and allowed to cool at room temperature for a period of 48 h.

Total arsenic was determined using atomic fluorescence equipment with a hydride generator (PS Analytical, PSA 10.055 Millenium Excalibur). The results were interpolated into a calibration curve (0.5 to 70 mg/L) from the ClinChek Urine Control Lyophilized for Trace Elements Level 10 × 10 mL reference standard (IRIS Technologies) in synthetic urine (concentration range of 32–51 µg/L). 

Drinking water: Water from the municipal capital of Peñón Blanco, Durango, was analyzed, including the metropolitan area and regions surrounding houses (tap water and purified water, if used). Sampling was carried out in duplicate, 15 days apart. To each 12 mL water sample, 0.5 mL of KI at 50% (*m/v*) in 10% ascorbic acid and 7.5 mL of concentrated hydrochloric acid were added and they were left to stand for 30 min at room temperature, stirring continuously. They were placed in the spectrometer for atomic fluorescence reading with a hydride generator. The concentration of As was determined using a National Institute of Standards and Technology (NIST) reference concentration of 60.45 ± 0.72 µg/L. The blank for analyzing the water samples was tri-distilled water, and the urine blank was synthetic without added arsenic. Previous to the determination in the test samples, the method was standardized for determining As in water and urine.

Water: the linearity was determined in a range from 1 to 70 µg/L (r = 0.9993), precision at %CV = 7.91, accuracy at %DR = −2.52%, and limit of quantification at 1 µg/L. 

Urine: the linearity was determined in a range from 0.5 to 70 µg/L (r = 0.9996), precision at %CV = 5.43, accuracy at %DR = 10.54, and the limit of quantification at 0.5 µg/L (r = 0.9996).

#### 2.4.2. Determination of Pioglitazone in Human Plasma

Blood sample: For pharmacokinetics, 3 mL samples were taken from individuals before fasting, with no citrus consumption (3 days before). Pioglitazone 30 mg p.o. was administered according to the sampling profile (0.5, 1, 1.5, 1.5, 2, 2.5, 3, 3.5, 4, 4.5, 5, 6, 9, 12, 24, and 36 h). Each sample was centrifuged, and the plasma was stored at −70 °C until analysis. A liquid–liquid method with ethyl acetate was used to extract pioglitazone from the plasm. Throughout a validated chromatographic method (NOM-177SSA1-2013) the molecule of pioglitazone was identified in an Agilent 1100 series HPLC under the following conditions: mobile phase water/acetonitrile (50:50, *v/v*) at pH = 3.1, injection volume 100 µL, flow rate 1.2 mL/min, stationary phase in a C18 column, 4.6 × 100 mm, 3.5 µm, temperature 22 °C, detection at 269 nm. The concentration of pioglitazone was in the range of 30 to 2000 ng/mL. After single compartment analysis, elimination constant (Kel), half-life time (t½), area under the curve from 0 to infinity (ABC0∞; trapezoid method), volume of distribution (VD), clearance (Cl), and pharmacokinetic parameters were obtained from the time course. Pharmacokinetic parameters were calculated by non-compartment analysis using Origin Pro 2018 software.

#### 2.4.3. In Silico Analysis of Interaction of Isoenzymes CYP 3A4 and CYP 2C8 with As

The sequences of both enzymes were obtained from the NCBI nucleotide database (http://www.ncbi.nlm.nih.gov/guide/genes-expression/ (accessed on 15 January 2018)). CYP 3A4 has 34,205 base pairs, and CYP 2C8 has 39,726 base pairs in its specific gene sequence. 

For the crystallographic structures of the sequences, the free program PyMol version 1.7, Schrodinger Sales Center, installed on an ICORE 5 desktop computer, was used.

The density functional theory (DFT) analysis of the isoforms was based on an in silico study using a mathematical algorithm on a five-core computer for fast convergence, determining the structure with the lowest energy.

#### 2.4.4. Statistical Analyses

Statistical analysis was performed using the Origin Pro 2018 package.Anthropometric parameters were compared by Mann–Whitney U test (*p* ≤ 0.05).For pioglitazone pharmacokinetics, Shapiro–Wilk analysis was used (n ≤ 3): *p* > 0.05, *t*-Student (independent group comparison).Arsenic interactions and anthropometric and clinical exposure parameters with pioglitazone pharmacokinetics were determined by Spearman correlation.

## 3. Results

The median value for the T2DM group in relation to age was of 56 years, weight of 75 kg, and height of 1.6 m. The healthy group had a median age of 44 years, weight of 75 kg, and height of 1.6 m (Table 1).

The clinical parameters analyzed showed that the diastolic blood pressure, cardiac frequency, fasting glucose, HbAc1α, insulin, TGO, and TGP values in the T2DM group are almost twice as high as the healthy group; *p* ≤ 0.05; Table 1.

The As concentration analysis in urine showed 15.65 ± 3.5 ppb for the T2DM group and 8.75 ± 3.11 ppb for the healthy group, with a 1:1.8 ratio, indicating the higher value of the T2DM group. However, these values do not reach statistical difference (*p* ≤ 0.05); the T2DM group is above 10 ppb, considered the maximum limit value for As in urine, and the healthy group is within the range, as shown in Figure 1.

Thirty-six samples of water for human consumption were obtained in duplicate 5.5%, with values ≤ 0.025 mg/L (limit allowed by NOM-127-SSA1-2021). Samples 10 and 39 showed minimum values of 0.019 mg/L. On the other hand, 94.5% gave values of 2.41–877.08% above the permitted limit; Table 2.

The pharmacokinetics of pioglitazone was analyzed at a dose of 30 mg p.o. This analysis was conducted under moderate to severe risk of contamination in the population using water contaminated by As. Figure 1 shows the time course constructed for the healthy and T2DM groups, which shows a slight shift towards higher values in the area under the curve (T2DM; light green line) concerning group 1 (healthy; dark green line); see Figure 2.

The data were corroborated by analyzing the population pharmacokinetic parameters, which showed significant differences in ABCt, which was 0.26 times higher in the T2DM group (*p* = 0.0312), while Ka was 3.7824 times higher (*p* = 0.000023) and Cl was 0.781 (*p* = 0.0015) times higher in the healthy group vs. T2DM group, as shown in Table 3.

The theoretical in silico analysis showed that the catalytic sites for CYP 3A4 and CYP 2C8 are identical in base pairs (100% homology). In comparison, the surrounding regions showed 50% homology in CYP 3A4 and CY2C8 (conserved residues in both isoenzymes: RIWRAGTPFSRCGA),. Figure 3 shows the CYP 3A4 structure. The effects of As on the Fe (heme group of CYP 3A4) were calculated using the DFT methodology. The heme group, cysteine 442 (C442), and the Fe core were considered the first step in the molecular model, Figure 3a. Figure 3b shows the atomic structure; after that, we added the corresponding hydrogen to complete the structure. Figure 3c displays our molecular model, which consists of a total of 93 atoms. This model is the base structure for structural optimization and load density calculations.

In the model, an As atom was placed 4 Å above the Fe nucleus to analyze the effects of As. Constrained optimization of the movement of the molecular fragment of residue C442, Fe, and As was performed. The other atoms were considered without movement or constant. It is possible to visualize a displacement of Fe towards N18, modifying the distance from 2.07 to 1.98 Å, and As is positioned between Fe and N50, the most negatively charged nitrogen in the control structure. An Fe atom is displaced from its original position as the distance to the S of C442 increases from 2.11 to 2.23 Å. These data show Fe destabilization by the presence of As, Figure 4, which probably affects its metabolism.

## 4. Discussion

The town of Peñón Blanco is located towards the center of the state of Durango; it is an area with many aquifers that provide water to the inhabitants for consumption and economic activities such as agriculture, livestock farming, and mining [31]. Peñón Blanco borders the area known as the “Laguna Region” where there is known to be a high level of arsenic contamination in drinking water of > 1100 µg/L [27]. Analysis of anthropometric and clinical parameters showed differences in fasting glucose (~89 mg/dL), insulin (~2 IU/mL), TGO (~22 IU/mL), and TGP (~12 IU/mL), being higher in the T2DM group vs. healthy group. Differences were also found in systolic pressure and heart rate, reflecting the expected deteriorating condition in T2DM patients. 

Concerning As exposure, in previous work in the town of Peñón Blanco, Dgo, As concentrations were quantified in drinking water and the results showed that As concentrations have values of 0.0815 ± 0.05 mg/L, classified by the Health Secretariat as a high exposure concentration (0.0251–0.150 mg/L). The above agrees with González-Coronado in 2007, who reported 0.010 mg/L [32]. In this sense, the values of Peñon Blanco water exceed the limit permitted, according to the Ministry of Health (<0.025 mg/L) and WHO (<0.010 mg/L) [33].

In this work, the results of As in urine in healthy and diabetic populations showed no significant differences. However, the T2DM group presents a higher average value of As concentration in urine than the healthy group, which is consistent with higher exposure to As in drinking water for this group. Navas-Acien et al. in 2006 found that at concentrations of 0.007 ppm of arsenic in the urine, there is a significant relationship between a population exposed to As (high concentrations) and development of T2DM [34]. Therefore, there is a risk of developing T2DM (healthy group) or the alteration of the metabolic process in the liver. 

The pharmacokinetic parameters for each population showed significant differences in absorption constant (Ka), Cl, and area under the total curve (ABCt). ABCt is an indirect marker of drug bioavailability [35], so the value suggests a variation in the T2DM group compared to the healthy group. This variation is relevant for the glycemic control of the T2DM group.

Exposure is one factor that may explain the variation in pharmacokinetic parameters. This study’s results are not statistically significant; however, exposure is high in the whole population. In this context, a deficient metabolizing function of cytochrome P450 enzymes has been documented [30] as a phenomenon associated with exposure to high concentrations of As in drinking water [34].

Drug metabolization deficiency and pharmacokinetic alteration affect treatment outcomes. This could be explained by the decrease in function of the CYP 3A4 and CYP 2D6 isoenzymes. The primary drug used to treat T2DM is pioglitazone. It is used for glycemic control, with 61% metabolized by CYP 3A4 and 39% by CYP 2C8. Therefore, the higher bioavailability in the T2DM group compared to the healthy group is explained. The analysis of Ka, Cl, and ABCt did not reveal statistically significant differences involving As in this variation (correlation analysis). However, the trend indicates a clear risk-explained association based on the following considerations:

The average age of populations is different, with the T2DM group being older than the healthy group and the older age group having higher As exposure.Due to the associations previously established in the literature that refer to the fact that chronic exposure to high concentrations of As decreases the expression of cytochrome P450, the T2DM group tends to metabolize the drug slowly compared to the control population.The presence of As in the cytochrome P450 structure could generate variation in pharmacokinetic parameters, thus, the ABCt is higher in the TDM2 group compared to the healthy group. In the absence of sufficient cytochrome action, the drug seems more bioavailable in systemic circulation for a longer time, thus causing the difference in the populations.

Supporting the previous assertion in the in silico analysis, we reflect that the charge density for the heme group iron of CYP 3A4 and CYP 2D6 isoenzymes is inhibited in the charge transfer capacity when As is present. Thus, the most significant charge density changes are in the sulfur of C442 and As. As absorbs or gives up charge from the system by displacing iron from the central site of the heme group in the enzyme, altering its metabolic function [35,36]. The effects of arsenic, particularly carcinogenic in humans, have been extensively studied, and its mechanism of action at the molecular level has been determined. However, studies by several authors, including Naraharisseti et al., have yielded data suggesting the importance of arsenic as a negative regulator of the expression of some cytochrome P450 isoforms [30]. Noreault et al., in 2005, found that at exposure concentrations higher than 0.005 ppm of arsenic in urine, DMA (dimethyl arsenic) and MMA (monomethyl arsenic) species bind to nuclear receptors, decreasing the expression of CYP 3A4 and its metabolizing ability [37]. When pioglitazone is administered to patients exposed to As, it only metabolizes 61%. For that reason, alteration of bioavailability is expected in the T2DM group.

On the other hand, the metabolic function of the CIT P450 isoforms may be due to other circumstances, such as a probable occurrence of polymorphism in either CYP 3A4 or CYP 2C8. Yamada et al. analyzed a probable polymorphism in CYP 3A4 (SNP 13989 A/G Ile 118 Val) [38,39]. They found no conclusive results that could relate to variation in pioglitazone pharmacokinetics. Likewise, Hart and Zhong, in 2008, indicated that there is less than a 1% probability of finding a polymorphism in these isoenzymes [40].

## 5. Conclusions

The results suggest that As modifies the enzymatic function of CYP 3 A4 and CYP 2C8 isoforms, leading to the altered metabolic function of CYP 2D6 and CYP 3A4 in patients with T2DM. Consequently, metabolism variation modifies the bioavailability of pioglitazone and, therefore, the expected final effect.

Likewise, considering the levels of As found in clinically healthy populations and agreement with the results of other authors, a long-term T2DM picture could be triggered in this population, depending on the continuous exposure.

## Figures and Tables

**Figure 1 ijerph-20-01901-f001:**
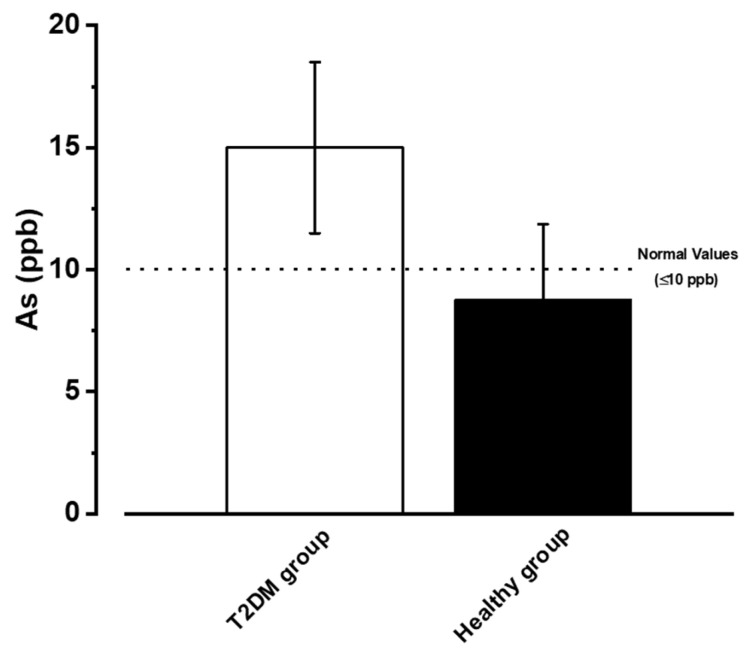
Urinary As levels in the participating groups, the graph represents the X ± E.E. of the urinary As concentration (ppb).

**Figure 2 ijerph-20-01901-f002:**
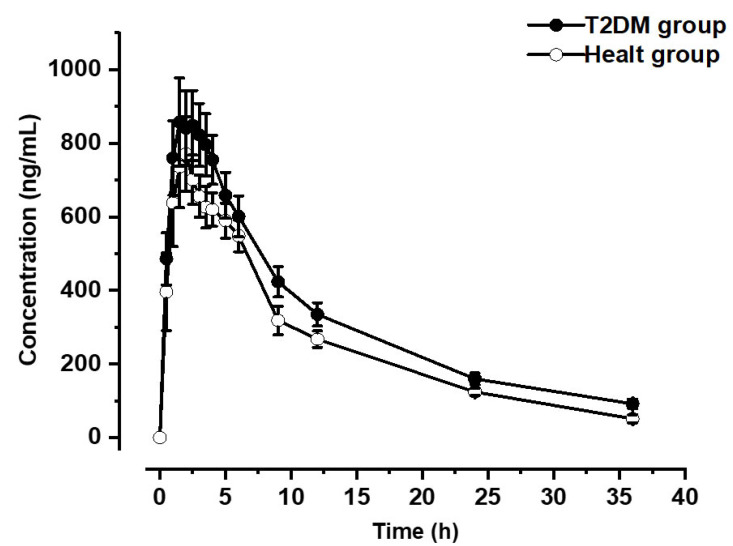
Concentration vs. time curve for control and case populations. Each line represents the X ± E.E. of the plasma concentration (ng/mL) of pioglitazone, after oral administration of a 30 mg tablet of pioglitazone.

**Figure 3 ijerph-20-01901-f003:**
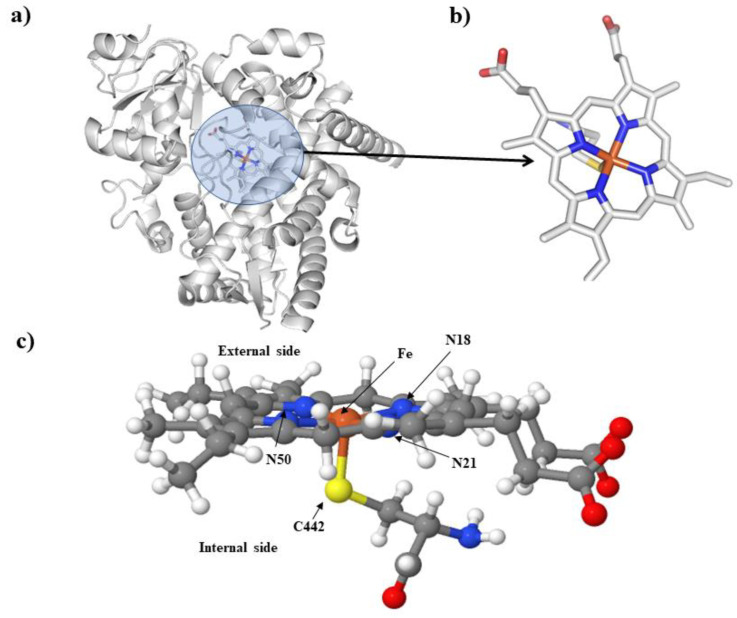
(**a**) Structure of CYP 3A4 obtained by homology of crystal 4KW9, (**b**) heme group’s atoms, Fe nucleus, and the amino acid C442, all extracted from the structure of CYP 3A4, and (**c**) again, the heme group with Fe and C442.

**Figure 4 ijerph-20-01901-f004:**
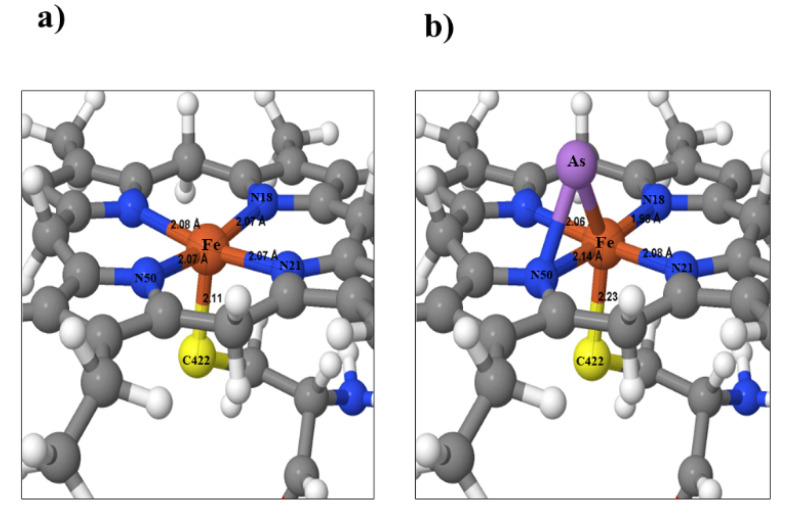
Close-up view of the Fe nucleus of the optimized form structures of the heme group atoms. (**a**) The control structure, (**b**) the molecular structure with As.

**Table 1 ijerph-20-01901-t001:** Clinical and anthropometric data of the study population representing the median (Xmin–Xmax).

Parameter	Healthy Group	T2DM Group
Males (*n* ^1^)	1	9
Females (*n*)	10	11
	Median (Xmin ^3^–Xmax ^2^)	Median (Xmin–Xmax)
Age (years)	44 (29–84)	56 (33–71)
Weight (kg)	75 (55–92)	75 (53–104)
Height (meters)	1.6 (1.52–1.66)	1.6 (0.94–1.75)
Systolic blood pressure (mmHg)	110 (100–150)	130 (100–188) *
Diastolic blood pressure (mmHg)	80 (60–100)	80 (50–120)
Cardiac frequency (beats per minute, bpm)	78 (68–92)	72 (59–92) *
Fasting glucose (mg/dL)	85 (68–95)	174 (101–322) *
HbAc1α (%)	5.5 (4.8–13.1)	9 (6–12.5) *
Insulin (IU/mL)	9.09 (2098–21.4)	5.55 (2–12.9) *
TGO (UI/mL)	24 (12–37)	41.2 (19.3–89.3) *
TGP (UI/mL)	16 (10–52)	31 (14–69) *

^1^ Number of individuals; ^2^ maximum value; ^3^ minimum value, * *p* ≤0.05 (Mann–Whitney U).

**Table 2 ijerph-20-01901-t002:** Data obtained from the sampled water intakes, showing the coordinates, the average concentration of the first and second sampling, the average, and whether it exceeds the values according to NOM-127-SSA1-2021 in Peñón Blanco, Durango, Mx.

	Coordinates		As mg/L		Average As mg/L	% Above the Limit NOM-127- SSA1-2021
	Latitude (N)	Longitude (O)	1st Sampling	2nd Sampling		
1	24°47.611′	104°1.511′	0.058	0.036	0.047	89.65%
2	24°47.246′	104°2.367′	0.074	0.071	0.073	193.16%
3	24°47.425′	104°2.229′	MNE	0.071	0.071	184%
4	24°47.529′	104°2.059′	0.046	0.074	0.061	141.67%
5	24°47.244′	104°2.098′	0.223	0.074	0.149	497.03%
6	24°48.013′	104°1.273′	0.066	0.00022	0.033	33.67%
7	24°47.941′	104°2.372′	0.034	0.038	0.036	44.85%
8	24°47.281′	104°1.855′	0.279	0.00022	0.139	459.39%
9	24°47.914′	104°1.349′	0.096	0.079	0.087	251.88%
10	24°48.448′	104°1.963′	MNE	0.019	0.019	Acceptable
11	24°47.910′	104°1.439′	0.083	0.084	0.084	236.09%
12	24°47.324′	104°2.026′	0.00022	0.078	0.039	58.33%
13	24°47.733′	104°1.667′	0.074	0.072	0.073	194.13%
14	---	---	0.287	---	0.211	747.06%
15	24°48.208′	104°1.460′	0.043	MNE	0.043	72%
16	24°47.283′	104°2.395′	0.074	0.1004	0.087	250.69%
17	24°47.569′	104°2.121′	0.068	0.095	0.082	228.01%
18	24°47.733′	104°1.667′	0.066	0.00022	0.033	32.55%
19	24°48.157′	104°1.428′	0.082	0.0805	0.081	226.34%
20	24°47.233′	104°2.282′	0.072	0.068	0.071	182.43%
21	24°47.596′	104°2.342′	0.00022	0.096	0.048	92.46%
22	24°43.940′	104°5.330′	0.058	0.102	0.081	221.96%
23	24°47.800′	104°2.290′	0.027	0.095	0.061	146.35%
24	24°47.809′	104°2.027′	0.071	0.092	0.081	226.21%
25	24°47.727′	104°1.677′	0.065	0.0804	0.073	192.16%
26	24°47.417′	104°2.200′	0.067	0.086	0.076	207.48%
27	24°47.429′	104°2.245′	0.084	0.083	0.084	236.82%
28	24°48.030′	104°1.319′	0.044	0.063	0.054	116.73%
29	24°47.348′	104°2.272′	0.231	0.247	0.239	857.49%
30	24°47.769′	104°1.516′	0.064	0.062	0.063	155.44%
31	24°47.266′	104°2.402′	0.024	0.026	0.025	2.41%
32	24°47.700′	104°1.695′	0.049	0.053	0.051	107.51%
33	24°47.725′	104°1.716′	0.062	0.0604	0.061	145.38%
34	24°46.848′	104°1.705′	0.041	0.046	0.043	73.92%
35	24°47.010′	104°1.939′	0.219	0.269	0.244	877.08%
36	24°46.779′	104°2.883′	0.016	0.022	0.019	Acceptable

**Table 3 ijerph-20-01901-t003:** Pharmacokinetic parameters of the studied populations.

Population	^1^ Ke (1/h)	^2^ T ½ (h)	^3^ ABCt (ng.h/mL) *	^4^ Cmax (ng)	^5^ Tmax (h)	^6^ Ka (1/h) *	^7^ Vd (L)	^8^ Cl (L/h) *
Diabetes	0.066 ± 0.012	15.22 ± 2.005	13747.49 ± 733.667	1066.594 ± 121.964	2.15 ± 0.254	0.501 ± 0.116	39.202 ± 4.5438	1.859 ± 0.106
Controls	0.089 ± 0.017	10.56 ± 1.657	10909.6 ± 1039.06	883.081 ± 87.9569	2 ± 0.387	2.396 ± 0.467	37.810 ± 5.234	2.640 ± 0.232

^1^ Ke elimination constant; ^2^ half-life time; ^3^ area under the total curve; ^4^ maximum concentration; ^5^ maximum time; ^6^ absorption constant; ^7^ absorption volume; ^8^ clearance; *t*-Student: * *p* ≤ 0.05.

## Data Availability

Not applicable.

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
