# Peer review of "Arsenic: A Perspective on Its Effect on Pioglitazone Bioavailability"

_ijerph, 2023, doi:10.3390/ijerph20031901_

Round 1

Reviewer 1 Report

The article is interesting and has a clear approach to the subject, however some suggestions were made with the aim of positively impacting the quality of the manuscript.

Methodology: Data related to As analysis: more detailed information about the quantification protocol is missing, as well as recovery data, LOD and LOQ.

A section on the statistical approach would be important.

A multiparametric statistical approach would be important to establish correlations between the results found, such as As concentration, biometric variables and the health status of the population.

Results and discussions: thinking about a possible influence of the gender factor, it would be important for the research to obtain an equal number of female and male individuals.

More current references can be used to support sentences.

Author Response

I am grateful for the review of the paper, in the attached document I send the answers to the comments.

Reviewer 2 Report

The manuscript addresses a topic of potential interest related to the pharmacokinetics of an antidiabetic drug and moderate to high exposure to inorganic arsenic through contaminated water.

The authors compared the pharmacokinetics of a 30 mg oral dose of pioglitazone between two groups exposed to arsenic, one apparently healthy and the other with type 2 diabetes mellitus (T2DM). The results show a lower clearance of pioglitazone that favors its greater bioavailability in individuals with T2DM. However, this alteration in the pharmacokinetics of the antidiabetic, aparently was not related to the internal arsenic dosimetry of the participants in an endemic area of Durango, Mexico. 

In addition, the authors carried out in-silico analysis, proposing the charge density for the heme group iron of CYP 3A4 and CYP 2D6 isoenzymes is inhibited in the charge transfer capacity when As is present, displacing iron from the central site of the heme group in the enzyme altering its metabolic function of CYPs.

Below I list my observations:

1-Title of the manuscript. – The tittle does not agree with the finding found. The changes in the pharmacokinetic parameters are consistent with the metabolic change itself in T2DM.

2-Ln25.-Results in abstract. Clarify that arsenic exposure in Healthy group and T2DM group were exposed to drinking water higher to 0.010 ppm.

3-Ln 78- However, contaminants such arsenic, at concentrations of 2.5-5 mM, favor a decrease in CYP 3A4 expression. Correct the concentration, 2.5-5 milimolar  is extraordinarily high for a cell to survive.

4-Ln 95- clarify that NIST 1643e is for arsenic in water.

5- Ln104-105-this paragraph is very confusing. Authors used a urine standard with a concentration as high as 1 ppm and add it to a urine mixture? whose were it? What concentration did the donors of the urine samples have? What final concentration did the urinary standard of arsenic have?...

5- Ln 108- A major limitation of the study was that there was no group that was not from the endemic area with arsenic water contaminated and that the two study groups were exposed to similar concentrations of arsenic. 

6- Ln133.The normal urine specific gravity is one, not zero. Ln 135-136 – Was any pretreatment performed on the urine samples prior to analysis? Were the participants asked not to eat seafood for a week prior to donating the urine sample for Total arsenic evaluation in urine?

7- Ln145-147- Clarify that NIST standard used was for trace elements in water and revise the range of arsenic concentrations of calibration graph (1 to 70  μg/mL is iqual to 1-70 mg/L).

8-Ln 159-161.  More information is needed on how the pharmacokinetic parameters were calculated; did you use any specialized software?

9- A description of the statistical methods used is needed.

10- Table 1. Revise units of cardiac frequency. Explain why used Chi2 for comparative test when data are expressed in mean +- error standard? 

11-Ln 182-183. The results of urinary total arsenic concentrations for more clarity should be expressed in parts per billion and not in ppm; that is, 15.65 ± 0.350 μg/L or ppb for the T2DM group and 8.75 ± 0.311 μg/L, for healthy group.   How are these urinary arsenic levels according to the reference values? It is noteworthy that with the limited group of participants for each group they can use a parametric statistic and express their results as mean +- standard error. It would be desirable to visualize the comparative concentrations between the two groups in box plot type graphs and indicate what statistic test used for (*p≤ 0.05) reported.

12- Authors must refer to NOM-127-SSA1-2021 instead of NOM-127-SSA1-1994 where the limit for arsenic in water was 0.050 ppm. The Mexican normative NOM-127-SSA1-2021 indicate limit of arsenic in drinking water of 0.025 ppm and future reduction to 0.010 ppm. Thus, authors could include in discussion this information.  In addition, it would be interesting for the authors to comment if participants who consuming water with concentrations greater than 0.100 ppm presented higher urinary concentrations of total arsenic.

 13-Figure 2. Units of pioglitazone, in y axis is wrong. Indicate for table 3 the statistics test  used for comparative analysis and indicate if pharmacokinetics values were adjusted to normal distribution.

14- Ln 236- “Laguna Region” use capital letter.

15-Some references are not listed as : Benítez-Leal et al. (2009);  González-Coronado  et al (2007),  Herrera-Saucedo et al (2010), Hart and Zhong (2008)  while other are incomplete: Naraharisseti et al.(27), Noreault et al.,(26) and Navas Acien et al (30).

16- It is quite confusing that in ln 183 expressed differences between urinary arsenic levels for healthy and T2DM groups while in In 248 indicated no significant difference, is imperative that this information be clarified. 

17. Some grammatical mistakes must be corrected.

Author Response

(The authors gave the same response as above.)

Round 2

Reviewer 2 Report

Unfortunately, not all the previous observations were satisfactorily addressed and several mistakes were made in the additional information that do not help to improve the quality of the information. Authors should review the following

Ln 24. Change sentence to read: The healthy and T2DM groups were exposed via drinking water to >0.010 ppm.

Ln 78 Note that reference Toxicol. Appl. Pharmacol. 2005, 2, 174-182. https://doi.org/10.1016/j.taap.2005.04.008, refer to 2-5 µM, not mM. Also, note that reference 33 does not correspond to Noreault,  et al 2005. Since the authors  made many changes to the bibliography, it would be wise to double-check in the text & references.

Ln96-96 – Change the sentence to read:…… (NIST; water standard 60.45 ± 0.72 µgAs/L), reference standard ClinChelk- Control Urine, Control Lyophilized for Trace Elements Level (IRIS; urine standard 32-51 µgAs/L) .

Ln 99-131- The "Preparation of Solutions" section  is not necessary, I suggest that it be eliminated or reported in supplementary information. However, it is necessary to clarify why urine pools were performed for the determination of Total arsenic and plasma pools for the determination of pioglizatone.

 A major concern is the precision and accuracy of arsenic concentrations determination, particularly in urine samples. It is very important that the authors know that inorganic arsenic and its metabolites (mono and dimethylated) are normally present in human urine samples, the latter must be converted to inorganic arsenic using a process of mineralization of the samples at high temperatures, more than 80 oC during long time. 

Note taht is wrong the units in line 174: ….reference concentration 60.45 ± 0.2 µg/L not mg/L. Also in Ln 183, accuracy and precision were not reported and detection limit for arsenic ub drinking water is wrong. It is not possible that detection limit for arsenic in water was 1ppm, but results in table 2 were much lower than 1ppm.

Table 1 now reports the results in median (range) but the statistical test to evaluate the differences should be Mann Whitney U test not  Chi2  due to Unfortunately, not all the previous observations were satisfactorily addressed and several mistakeswere made in the additional information that do not help to improve the quality of the information. Authors should review the following

Ln 24. Change sentence to read: The healthy and T2DM groups were exposed via drinking water to >0.010 ppm.

Ln 78 Note that reference Toxicol. Appl. Pharmacol. 2005, 2, 174-182. https://doi.org/10.1016/j.taap.2005.04.008, refer to 2-5 µM, not mM. Also, note that reference 33 does not correspond to Noreault, T.L et al 2005. Since the authors  made many changes to the bibliography, it would be wise to double-check in the text & references.

Ln96-96 – Change the sentence to read:…… (NIST; water standard 60.45 ± 0.72 µgAs/L), reference standard ClinChelk- Control Urine, Control Lyophilized for Trace Elements Level (IRIS; urine standard 32-51 µgAs/L) .

Ln 99-131- The solution preparation part is not necessary, I suggest that it be eliminated or reported in supplementary information. However, it is necessary to clarify why urine pools were performed for the determination of total arsenic and plasma pools for the determination of pioglizatone.

 A major concern is the precision and accuracy of arsenic concentrations, particularly in urine samples. It is very important that the authors know that inorganic arsenic and its metabolites (mono and dimethylated) are normally present in human urine samples, the latter must be converted to inorganic arsenic (with a process of mineralization of the samples at high temperatures, more than 80 oC during long time ) for the determination of Total Arsenic in urine.

Note taht is wrong the units in line 174: ….reference concentration 60.45 ± 0.2 µg/L. Also in Ln 183, accuracy and precision were not reported and detection limits are wrong. It is not possible that detection limit for arsenic in water was1ppm, but results in table 2 were much lower than 1 ppm.

Table 1 now reports the results in median (range) but the statistical test to evaluate the differences should be Mann Whitney U not  Chi 2  due to it is to evaluate differences in proportions.

Change the figure caption of Figure 1 Urinary As levels in the participants groups, the graph represents the X ± E.E. of the urinary As concentration (ppb), p>0.05_(_t-student). Is not acceptable refers to experimental group to study participants.

Ln 408 the paragraph is not understood: The value exceeds the permitted limit, according to the Ministry of Health (<0.025 mg/L trend to decrease to <10 mg/L over the next six years) and the WHO (<0.0 10 mg /L)”.

Ln 411 this sentence is contradictory: no significant differences (p≤0.05). The value p≤0.05 refers to significant difference. 

Change the figure caption of Figure 1 Urinary As levels in the participants groups, the graph represents the X ± E.E. of the urinary As concentration (ppb), p>0.05(t-student test). Is not acceptable refers to experimental group to study participants.

Ln 408 the paragraph is not understood: The value exceeds the permitted limit, according to the Ministry of Health (<0.025 mg/L trend to decrease to <10 mg/L over the next six years) and the WHO (<0.0 10 mg /L)”.

Ln 411 this sentence is contradictory: no significant differences (p≤0.05). The value p≤0.05 refers to significant difference.

Author Response

Greetings, I appreciate the comments to the paper and I attach the responses to the comments derived from revision 1. 
Also, I comment that derived from the second review there are some comments that were duplicated by the reviewer so we only responded to one comment and the duplicate was not considered.

Thank you
